# Dynamic Monitoring of a Bridge from GNSS-RTK Sensor Using an Improved Hybrid Denoising Method

**DOI:** 10.3390/s25123723

**Published:** 2025-06-13

**Authors:** Chunbao Xiong, Zhi Shang, Meng Wang, Sida Lian

**Affiliations:** 1School of Civil Engineering, Tianjin University, Tianjin 300350, China; chunbao@tju.edu.cn; 2School of Civil Engineering, Hebei University of Engineering, Handan 056009, China; liansida@tju.edu.cn

**Keywords:** GNSS-RTK, improved denoising method, dynamic monitoring, bridge

## Abstract

This study focused on the monitoring of a bridge using the global navigation satellite system real-time kinematic (GNSS-RTK) sensor. An improved hybrid denoising method was developed to enhance the GNSS-RTK’s accuracy. The improved hybrid denoising method consists of the improved complete ensemble empirical mode decomposition with adaptive noise (ICEEMDAN), the detrended fluctuation analysis (DFA), and an improved wavelet threshold denoising method. The stability experiment demonstrated the superiority of the improved wavelet threshold denoising method in reducing the noise of the GNSS-RTK. A noisy simulation signal was created to assess the performance of the proposed method. Compared to the ICEEMDAN method and the CEEMDAN-WT method, the proposed method achieves lower RMSE and higher SNR. The signal obtained by the proposed method is similar to the original signal. Then, GNSS-RTK was used to monitor a bridge in maintenance and rehabilitation construction. The bridge monitoring experiment lasted for four hours. (Considering the space limitation of the article, only representative 600 s data is displayed in the paper.) The bridge is located in Tianjin, China. The original displacement ranges are −14.9~19.3 in the north–south direction; −26.9~24.7 in the east–west direction; and −46.7~52.3 in the vertical direction. The displacement ranges processed by the proposed method are −12.3~17.2 in the north–south direction; −24.6~24.1 in the east–west direction; and −46.7~51.1 in the vertical direction. The proposed method processed fewer displacements than the initial monitoring displacements. It indicates the proposed method reduces noise significantly when monitoring the bridge based on the GNSS-RTK sensor. The average sixth-order frequency from PSD is 1.0043 Hz. The difference between the PSD and FEA is only 0.99%. The sixth-order frequency from the PSD is similar to that from the FEA. The lower modes’ natural frequencies from the PSD are smaller than those from the FEA. It illustrates the fact that, during the repair process, the missing load-bearing rods made the bridge less stiff and strong. The smaller natural frequencies of the bridge, the complex construction environment, the diversity of workers’ operations, and some unforeseen circumstances occurring in the construction all bring risks to the safety of the bridge. We should pay more attention to the dynamic monitoring of the bridge during construction in order to understand the structural status in time to prevent accidents.

## 1. Introduction

Large-scale civil structures are influenced by ambient excitation, material aging, and structural creep [1]. When structural damage affects safe operations, some actions should be taken to avoid greater losses [2]. Considering sustainable development and economic conditions, maintenance and rehabilitation of structures are always a better choice when safety hazards arise in civil structures. To ensure construction safety, safety monitoring plays an important role in the maintenance and rehabilitation of construction. Safety monitoring can help construction managers anticipate construction risks and thus make more effective preventive decisions in advance. Compared to other monitoring technologies, such as automatic total stations [3] and accelerometers [4], GNSS-RTK has great potential for application in structural dynamic monitoring without climatic limitations, visibility requirements between stations, and errors caused by quadratic integration [5,6]. Multi-GNSS has been applied to bridge dynamic monitoring in previous studies [7,8,9,10]. However, GNSS-RTK accuracy is impacted by receiver measurement noise, multipath errors, etc. [11,12].

One of the methods to enhance the performance of GNSS-RTK for monitoring bridges was hardware modification [13]. However, hardware modification usually costs more than software innovation. Many filter algorithms have been developed to improve GNSS-RTK accuracy [14,15,16]. The EMD series decomposition methods (i.e., empirical mode decomposition, ensemble empirical mode decomposition, complete ensemble empirical mode decomposition with adaptive noise, and ICEEMDAN) are commonly utilized in noise reduction. The ICEEMDAN is the most ideal among the EMD series decomposition methods, which overcomes various shortcomings of other EMD series methods [17,18]. Some researchers have demonstrated that variational mode decomposition (VMD) is significantly better than other common signal decomposition methods [19,20]. However, the penalty factor and modal number have a significant impact on the decomposition effect of VMD. ICEEMDAN does not need to consider the penalty factor and modal number. Spectral selection, correlation coefficient, and normalized autocorrelation function are common methods for identifying noise [21,22]. DFA allows for evaluating a time-series long-range correlation, which provides a method to judge whether the time series contains noise [23]. In this study, we used DFA to identify whether the IMF decomposed from the GNSS-RTK monitoring signal contained noise. The traditional signal decomposition and noise judgment hybrid denoising method deletes the noisy IMF components. This way leads to the loss of some useful information. To avoid this problem, researchers utilized the wavelet threshold method to denoise the noisy IMF components and then reconstructed the resulting signal [24]. However, the traditional wavelet threshold denoising methods have disadvantages [25]. The signals obtained by the hard threshold denoising method still contain noticeable noise due to the hard threshold function being discontinuous. Soft threshold denoising suffers from distortion of the denoised signal caused by permanent bias. Considering these shortcomings, this study developed an improved wavelet threshold denoising method.

In this study, we propose an improved hybrid denoising method for bridge dynamic monitoring based on the GNSS-RTK sensor. The innovations and contributions are (1) The improved wavelet threshold function is superior to the traditional soft threshold function and hard threshold function. (2) The proposed method combining ICEEMDAN and the improved wavelet threshold denoising method is superior to the CEEMDAN-WT method and the traditional ICEEMDAN method. (3) ICEEMDAN is the most ideal among the EMD series decomposition methods. DFA can effectively estimate the correlation of non-stationary long-term time series. The improved wavelet denoising method is superior to the traditional wavelet denoising method in reducing the noise of GNSS-RTK. The proposed approach consists of the ICEEMDAN method, the DFA method, and the improved wavelet denoising method. The proposed approach is particularly suited to GNSS-RTK signal characteristics in comparison to commonly used denoising strategies. (4) Previous studies usually applied GNSS-RTK to monitor the bridge in its normal state. This study monitored the dynamic displacements of a bridge during maintenance and rehabilitation construction. The monitoring result revealed the characteristics of the bridge in maintenance and rehabilitation construction.

In the Section 2, ICEEMDAN, DFA, the improved wavelet threshold denoising method, and the proposed method are displayed. Section 3, first assesses the proposed method using a noisy simulation signal. Then, the stability experiment revealed the noise of GNSS-RTK and evaluated the performance of the improved wavelet threshold on the denoising RTK monitoring results. Next, the proposed GNSS-RTK-based method was implemented to monitor a bridge in maintenance and rehabilitation construction. Section 4 reveals the bridge’s features during maintenance and rehabilitation construction. The results also reminded us to pay more attention to the bridge in maintenance and rehabilitation construction to prevent safety problems caused by excessive changes in the mechanical system. Conclusions were summarized in theSection 5.

## 2. Methodology

### 2.1. ICEEMDAN

The ICEEMDAN method’s flow are1.Join $E_{k}\left[\omega^{\left(k\right)}\right]$ into s(t) to acquire $s^{\left(k\right)}(t)$:
(1)$$s^{\left(k\right)}(t)=s(t)+\beta_{0}E_{1}(\omega^{\left(k\right)}(t))$$2.$r_{1}(t)$ and $IMF_{1}(t)$ are calculated as follows:
(2)$$r_{1}(t)=\langle M(s^{\left(k\right)}(t))\rangle$$(3)$$IMF_{1}(t)=s(t)-r_{1}(t)$$3.Calculate $s^{\left(2\right)}(t)$:
(4)$$s^{\left(2\right)}(t)=r_{1}(t)+\beta_{1}E_{2}(\omega^{\left(k\right)}(t))$$4.$r_{2}(t)$ and $IMF_{2}(t)$ are acquired by repeating the second step:
(5)$$r_{2}(t)=\langle M(s^{\left(2\right)}(t))\rangle =\langle M(r_{1}(t)+\beta_{1}E_{2}(\omega^{\left(k\right)}(t)))\rangle$$(6)$$IMF_{2}(t)=r_{1}(t)-r_{2}(t)$$5.Acquire $r_{k}(t)$ and $IMF_{k}(t)$, for *k* = 3, …, *k*, as follows:(7)$$r_{k}(t)=\langle M(r_{k-1}(t)+\beta_{k-1}E_{k}(\omega^{\left(k\right)}(t)))\rangle$$(8)$$IMF_{k}(t)=r_{k-1}(t)-r_{k}(t)$$6.The preceding steps are repeated until the computation and decomposition are complete.

### 2.2. DFA

For a time-series x(i), the DFA steps are as follows:

1.Obtain y(k) by processing x(i) as Equation (9). Then, divide it into length n segments.
(9)$$\begin{array}{cc} y(k)=\sum_{i=1}^{k}[x(i)-\bar{x}], & k=1,2,\ldots ,N \end{array}$$2.To obtain local trends in each segment $y_{n}(i)$, nonlinear fitting between extreme points is accomplished using least squares and kth order polynomials:
(10)$$y_{n}(i)=\sum_{n=0}^{k}a_{n}i^{n}$$3.In each segment $y_{n}(i)$, remove the local trends, then average the squares of the outcome:
(11)$$F^{2}(n)=\frac{1}{n}\sum_{i=1}^{n}{\left[y(k)-y_{n}(i)\right]}^{2}$$4.Obtain the second-order fluctuation function:
(12)$$F_{q}(n)={\left[\frac{1}{N_{s}}\sum_{i=1}^{N_{s}}F^{2}(n)\right]}^{1/2}$$5.Change the length of the segments n, then repeat steps 2, 3, and 4. The curve of the full-sequence fluctuation function F(n) can be obtained as follows:
(13)$$F(n)\propto n^{\alpha }$$6.Calculate the scaling exponent α by $\frac{\log_{10}F(n)}{\log_{10}(n)}$

When α<0.5, noise takes up a significant portion of the signal. When α=0.5, the signal can be determined as white noise. When α>0.5, the signal is valid [26].

### 2.3. Improved Wavelet Threshold Denoising

The improved threshold function is(14)$$W(d,\lambda )=\left\{\begin{array}{cc} \mathrm{sgn}(d)\sqrt{{\left|d\right|}^{2}-{\left[\frac{2\lambda }{1+\exp(d-\lambda )}\right]}^{2}} & \left(\left|d\right|\geq \lambda \right)\\ 0 & \left(\left|d\right|<\lambda \right) \end{array}\right.$$(15)$$\lambda =\sigma \sqrt{2\ln L}$$(16)$$\sigma =\frac{\mathrm{median}(d)}{0.6745}$$

As shown in Equation (14), the improved threshold function is continuous at $\left|d\right|=\lambda$, which overcomes the drawback of hard threshold denoising.

When $\left|d\right|\geq \lambda$ and d→∞:(17)$$\underset{d\to \infty }{\lim}\frac{W(d,\lambda )}{d}=\underset{d\to \infty }{\lim}\frac{\mathrm{sgn}(d)\sqrt{{\left|d\right|}^{2}-{\left[\frac{2\lambda }{1+\exp(d-\lambda )}\right]}^{2}}}{d}\to 1$$

That is to say, when $\left|d\right|\geq \lambda$ and d→∞, the improved threshold function approaches the hard threshold function progressively. From this point, the improved wavelet threshold denoising overcomes the drawback of the soft threshold denoising. Figure 1 depicts the threshold function. As shown in Figure 1, the improved threshold function converges rapidly from the soft threshold function to the hard threshold function at $\left|d\right|\geq \lambda$. Meanwhile, the improved threshold function is continuous at $\left|d\right|=\lambda$.

### 2.4. The Proposed Method

This study proposed an improved hybrid denoising method based on the GNSS-RTK sensor. This method was applied to bridge dynamic monitoring. First, the GNSS-RTK monitoring signal was decomposed into IMF components by ICEEMDAN. Then, DFA was used to identify whether the IMF components contained noise. If the IMF component contained noise, reduce noise using the improved wavelet threshold denoising method. Finally, the denoised IMF components and the valid IMF components were combined to reconstruct the monitoring signal. Figure 2 displays the proposed method’s flow.

## 3. Experiments and Analysis

### 3.1. The Proposed Method Evaluation

Design the following signal:(18)$$y(t)=\underset{x(t)}{\underbrace{A_{1}\sin(2\pi f_{1}t)+A_{2}\cos(2\pi f_{2}t)}}+n_{1}(t)+n_{2}(t)$$
where $A_{1}=0.5$, $A_{2}=0.8$, $f_{1}=5$ Hz, $f_{2}=2$ Hz, $n_{1}(t)$ is the Gaussian noise, and $n_{2}(t)$ is a noise with the natural frequency of 0.04 Hz. x(t)=original signal, and y(t)=signal containing noise. The sampling time was 10 s, and the sampling frequency was 100 Hz. Figure 3a shows the simulation signals.

First, y(t) was decomposed into the IMF components and residual components using the ICEEMDAN method. Figure 3b displays the IMF components and a residual component. Then, DFA was used to identify the noisy component. As shown in Figure 3c, the indicator α of IMF1, IMF2, IMF3, IMF4, IMF5, and IMF6 is less than 0.5, and the others are larger than 0.5. It indicates that IMF1, IMF2, IMF3, IMF4, IMF5, and IMF6 contain noise, and others are valid components. A noisy signal was constructed with IMF1, IMF2, IMF3, IMF4, IMF5, and IMF6. Then, the noisy signal was denoised by the improved wavelet threshold denoising method. The signal obtained by the proposed method was constructed with the valid components and the denoised signal. In Table 1, the proposed method acquires lower RMSE and higher SNR than the ICEEMDAN method and the CEEMDAN-WT method. The signal obtained by the proposed method is more similar to the original signal x(t), as shown in Figure 3d. It indicates the superiority of the proposed method.

### 3.2. Stability Experiment

The GNSS-RTK stability experiment was conducted on stable ground near a river with two GNSS receivers. The plane positioning accuracy of the instruments was ±(1 cm + 1 ppm), and the elevation positioning accuracy was ±(2 cm + 1 ppm). This experiment applied GPS, GLONASS, and BDS. Figure 4 shows the rover station. Figure 5 shows the displacements, and the dashed lines mark the displacement range boundaries. The non-zero original displacements reveal the noise of GNSS-RTK. Figure 5d shows a lower displacement range than Figure 5b,c. It indicates that the improved threshold wavelet denoising method is superior to the traditional wavelet denoising methods in reducing GNSS-RTK noise.

### 3.3. Engineering Monitoring

The monitoring experiment was conducted on the Rainbow Bridge in Tianjin, China. The main bridge of the Rainbow Bridge is 504 m long and 29 m wide, and it is a concrete-filled steel tubular arch bridge. The rigid arch with a vector height of 32 m adopted a simple supported down-bearing flexible tie rod system. The Rainbow Bridge was opened to traffic in 1998 and was one of the top ten key projects in Tianjin at that time. Due to the high volume of traffic on the bridge, the bridge has suffered some degree of damage over years of use. According to the safety assessment, the bridge needed to be repaired with the replacement of all load-bearing rods. Figure 6a shows the Rainbow Bridge. The field experiment was conducted during the bridge maintenance and rehabilitation construction. The monitoring point was arranged in the middle of the bridge, which is the unfavorable cross-section. Considering the monitoring would not interfere with construction, the monitoring point was placed at the guardrail. Figure 6b shows the location of the monitoring point.

The monitoring experiment used two RTK receivers. At the monitoring point, one of the receivers served as the rover station. The other receiver was set up as the reference station in an open field about 200 m from the monitoring point. Figure 6g shows the reference station. The sampling frequency was 50 Hz, and the cutoff elevation angle was 15°. Figure 6c–f shows the instruments arranged at the monitoring point. The humidity was 28.7% RH, and the temperature was 24.31 °C. The wind direction was north–easterly, and the wind speed ranged from 0.00 to 2.55 m/s. The weather was fine.

Figure 7 and Table 2 show the displacements of the monitoring point. First, ICEEMDAN decomposed the original displacement into the IMF components and residual components. The standard deviation of the Gaussian noise affects the accuracy of signal decomposition. Setting too high may introduce too much noise, while setting too low may not have enough effect on the noise. Increasing the number of noise additions can improve the consistency of results, but it will increase computation time. If the number of iterations is too small, it may not be possible to obtain a good IMF, and if it is too large, it may lead to excessively long computation time. Considering the above, the parameters for ICEEMDAN were the standard deviation of the Gaussian noise was 0.2; the number of noise additions was 50; the maximum number of iterations was 100. Figure 8a displays the IMF components and residual components. Then, DFA was used to identify the noisy component. The size of the windows at which to evaluate the fluctuation was 50. As shown in Figure 8b, IMF2, IMF3, IMF4, IMF5, IMF6, IMF7, and IMF8 contain noise, where indicator α is less than 0.5. In addition, IMF1 in the north–south direction and east–west direction, as well as IMF9 in the east–west direction, contain noise, where the indicator α is also less than 0.5. The noisy signals were constructed using the IMF components containing noise. The noisy signals were reduced using the improved wavelet threshold denoising method. The wavelet type was db8, decomposition level was 4. As shown in Figure 8c, the improved wavelet threshold denoising method can effectively reduce the noise and retain useful information. At last, the signals were reconstructed. To obtain the north–south displacement signal, the denoised signal was recombined with IMF9, IMF10, IMF11, IMF12, IMF13, and the residual component. To obtain the east–west displacement signal, the denoised signal was recombined with IMF10, IMF11, IMF12, IMF13, and the residual component. To obtain the vertical displacement signal, the denoised signal was recombined with IMF1, IMF9, IMF10, IMF11, IMF12, and the residual component. As shown in Table 2, the proposed method produces a narrower displacement range compared to the original displacements. Figure 7 displays the displacement signal treated by the proposed approach, which reduced the noise while retaining the primary signal. It indicates the proposed method is efficient at reducing GNSS-RTK monitoring noise. In addition, Figure 7 and Table 2 show that the bridge’s vertical dynamic displacements were larger than its planar dynamic displacements.

## 4. Dynamic Characteristics Identification

The first six-order natural frequencies obtained by FEA (i.e., finite element analysis) are 0.1632 Hz, 0.6787 Hz, 0.8171 Hz, 0.8254 Hz, 0.9225 Hz, and 0.9945 Hz (The modal shapes from FEA were shown in Figure 9) [27]. Figure 10a shows the PSD (i.e., the power spectral density) of the monitoring results. The average sixth-order frequency from PSD is 1.0043 Hz. The difference between the PSD and FEA is only 0.99%. The sixth-order frequency from the PSD is similar to that from the FEA. However, the lower modes’ natural frequencies from the PSD are smaller than those from the FEA. The FEA was constructed based on the design of the bridge, assuming the bridge was in good condition. When the load-bearing rods are being replaced, although the bridge maintains the shape of an arch bridge, its structural mechanical system is different from that of the bridge in good condition. In good condition, all load-bearing rods of the bridge performed as they should. In maintenance and rehabilitation construction, some load-bearing rods are removed, and some load-bearing rods may have taken on more force than intended by the design. Thus, before the construction, the FEA for each stage of the repair process should be performed to determine the best repair solution. However, the FEM is only for key construction process nodes. The construction process is a continuous change process. The FEM is not able to comprehensively cover the entire continuously changing construction process. During the repair process, the missing load-bearing rods made the bridge less stiff and strong, which led to a lower natural frequency of the bridge. This is the reason why the lower modes’ natural frequencies from the PSD are smaller than those from the previous FEA. The external excitation frequency coinciding with the natural frequency of the bridge will cause the bridge to resonate. The resonance effect of the bridge may lead to the collapse of the bridge. The smaller natural frequency indicates that the bridge in maintenance and rehabilitation construction is less able than the bridge in good condition to withstand external incentives. The bridge still maintains the shape of an arch bridge in construction, so it preserves some mechanical characteristics of its design. This was shown by the fact that the sixth-order natural frequencies from the PSD and the FEA were similar in this study.

Figure 10b shows the construction site. Large construction machines worked on the bridge, some load-bearing rods were placed on the bridge, and workers were removing and installing load-bearing rods on the arches and decks of the bridge. These situations could not appear when the bridge is in normal condition. The complex construction situations significantly affect the bridge’s mechanical system. The process of removing and installing load-bearing rods is a process of destroying the bridge’s old load-bearing system and rebuilding a new load-bearing system. This process caused the bridge’s mechanical system to constantly change, and the bridge’s natural frequencies changed in tandem.

The smaller natural frequencies of the bridge, the complex construction environment, the diversity of workers’ operations, and some unforeseen circumstances occurring in the construction all bring risks to the safety of the bridge. We should pay more attention to the dynamic monitoring of the bridge during the construction in order to understand the structural status in time to prevent accidents.

## 5. Conclusions

This study monitored a bridge in maintenance and rehabilitation construction using the GNSS-RTK sensor, and an improved denoising method was developed to increase the GNSS-RTK’s accuracy. The proposed method includes (1) ICEEMDAN was used to decompose displacement signal into IMF components and the residual component; (2) use DFA to identify the noisy component; (3) to denoise the noisy components, the improved wavelet threshold denoising method was utilized; (4) reconstruct the displacement signal using denoised IMF components and valid components. The conclusions are
1.The improved wavelet threshold denoising method overcomes the drawbacks of the traditional wavelet threshold denoising method. The improved threshold function is continuous at $\left|d\right|=\lambda$, and rapidly from the soft threshold function tends to the hard threshold function at $\left|d\right|\geq \lambda$. The stability experiment also demonstrated that the improved wavelet threshold denoising method was more effective in reducing GNSS-RTK noise.The simulation experiment proved that the proposed method is superior to the ICEEMDAN method and the CEEMDAN-WT method. The proposed method acquired lower RMSE and higher SNR compared to the ICEEMDAN. The signal acquired using the proposed method is similar to the original signal.The bridge’s vertical dynamic displacements exceeded the planar dynamic displacements. The proposed approach processed fewer displacements than the initial monitoring displacements. It indicates the proposed method reduces noise significantly when monitoring the bridge based on the GNSS-RTK sensor.Dynamic characteristics identification revealed the bridge’s features during maintenance and rehabilitation construction. The sixth-order frequency from the PSD is similar to that from the FEA. The lower modes’ natural frequencies from the PSD are smaller than those from the FEA. It illustrates the features of the natural frequencies in the repair of bridges. During the repair process, the missing load-bearing rods made the bridge less stiff and strong, which led to the lower natural frequencies of the bridge being smaller. The bridge still maintains the shape of an arch bridge in construction, so it preserves some mechanical characteristics of its design, that the sixth-order natural frequencies from the PSD and the FEA are similar.The external excitation frequency coinciding with the natural frequency of the bridge will cause the bridge to resonate. The resonance effect of the bridge may lead to the collapse of the bridge. The smaller natural frequencies of the bridge, the complex construction environment, the diversity of workers’ operations, and some unforeseen circumstances occurring in the construction all bring risks to the safety of the bridge. We should pay more attention to the dynamic monitoring of the bridge during the construction, in order to understand the structural status in time to prevent accidents.

## Figures and Tables

**Figure 1 sensors-25-03723-f001:**
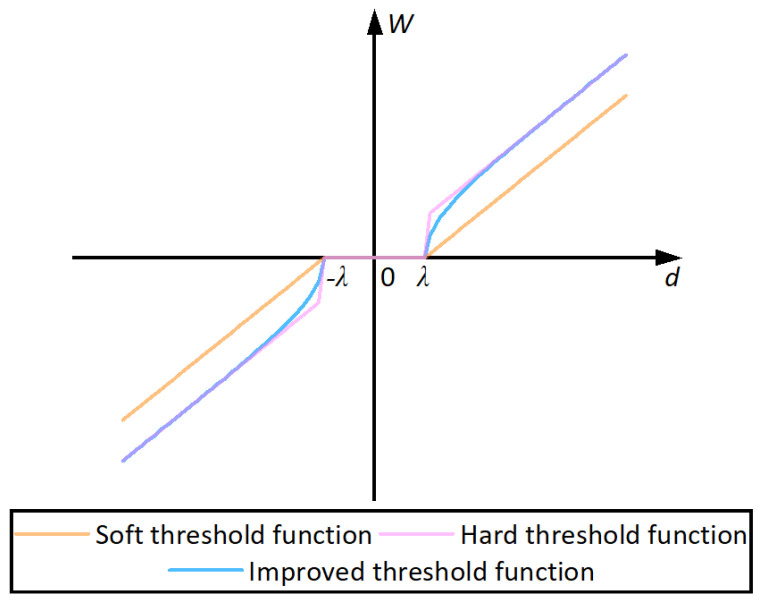
Threshold function.

**Figure 2 sensors-25-03723-f002:**
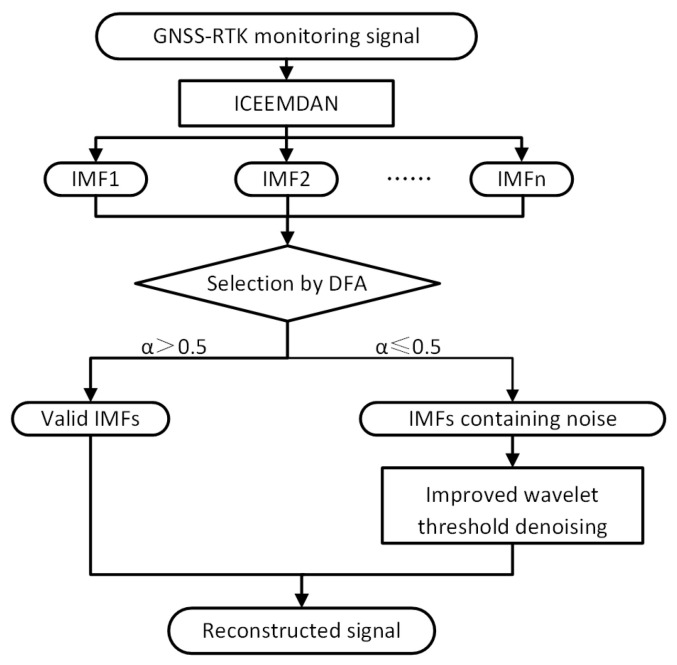
Flowchart of the proposed method. (Arrows indicate the direction and flow of the method.)

**Figure 3 sensors-25-03723-f003:**
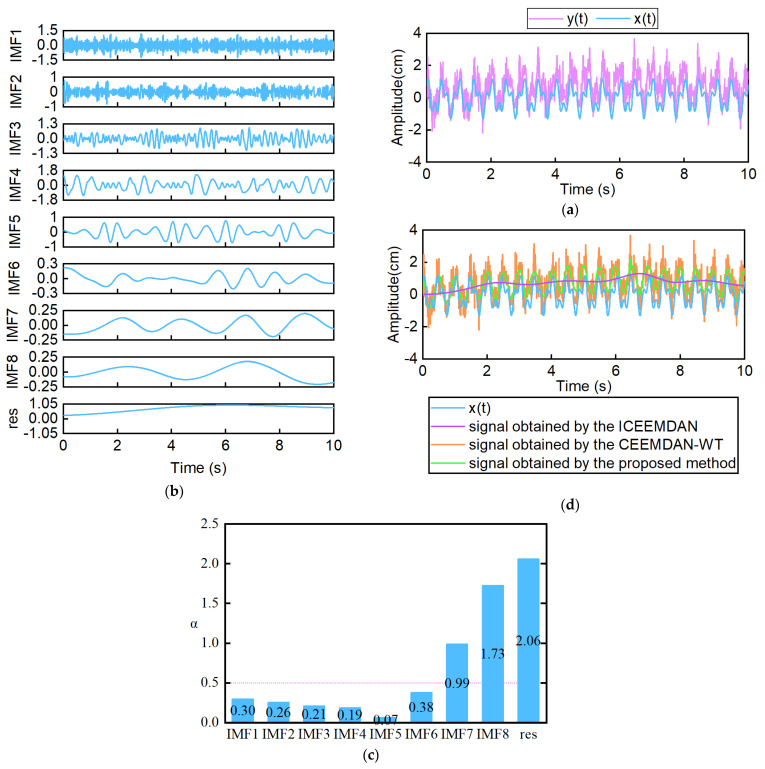
Performance evaluation of the proposed method. (**a**) Amplitudes of the original signal and the signal containing noise. (**b**) IMF components and residual components obtained by ICEEMDAN. (**c**) DFA result. (**d**) Amplitudes of the original signal and the signal obtained by filter.

**Figure 4 sensors-25-03723-f004:**
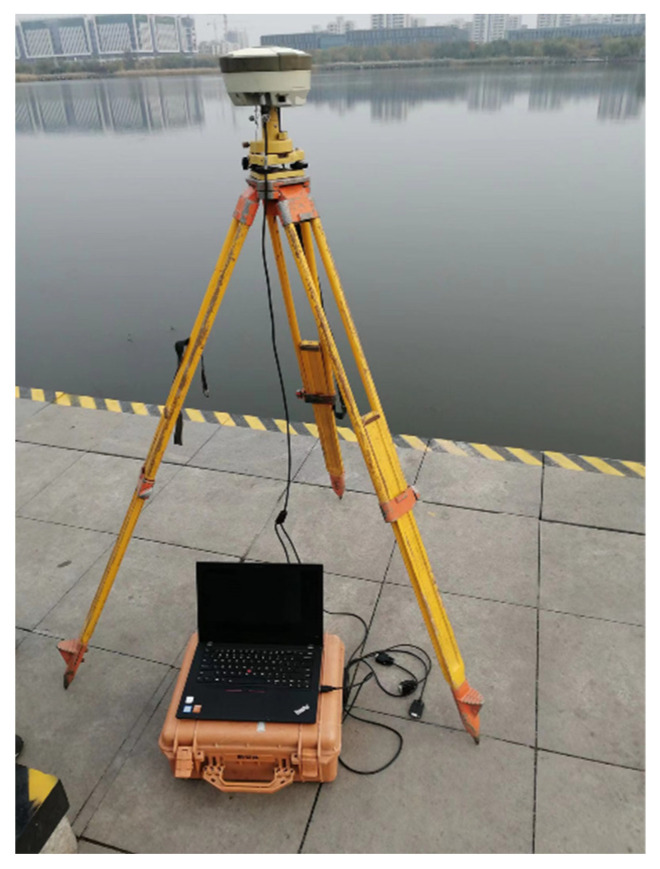
Rover station in the stability experiment.

**Figure 5 sensors-25-03723-f005:**
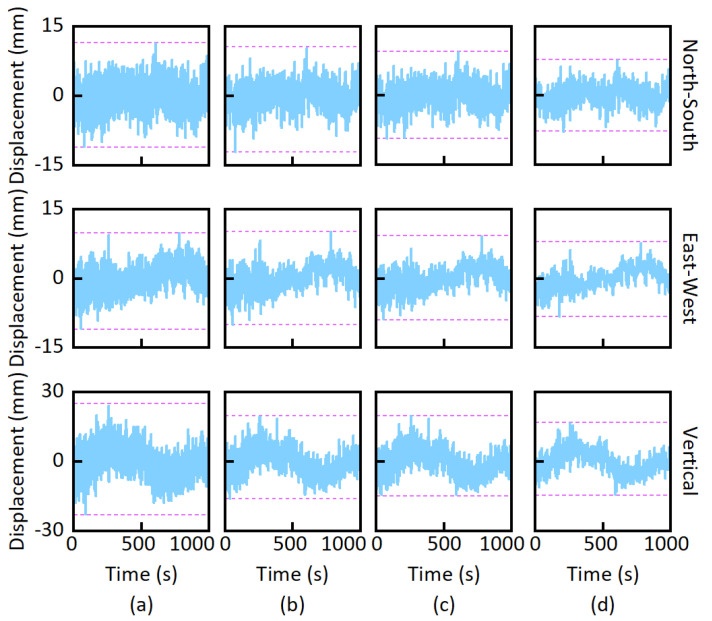
Displacements in the stability experiment. (**a**) Original displacements; (**b**) displacements obtained by the hard threshold wavelet denoising method; (**c**) displacements obtained by the soft threshold wavelet denoising method; (**d**) displacements obtained by the improved threshold wavelet denoising method. (Dotted lines mark the boundary of the displacements).

**Figure 6 sensors-25-03723-f006:**
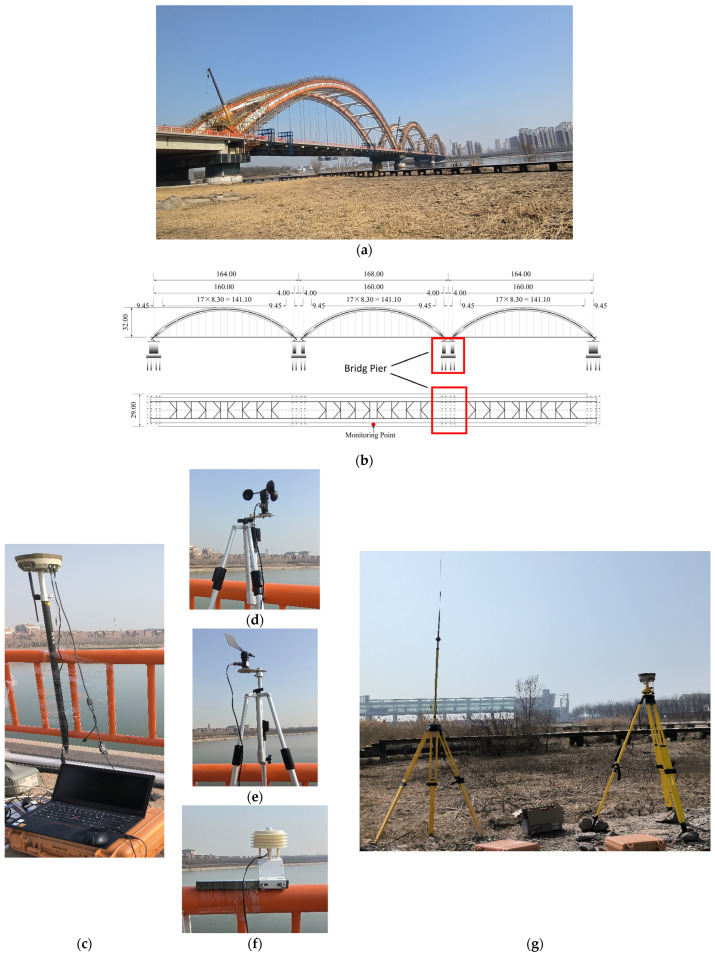
Rainbow Bridge. (**a**) Photo of Rainbow Bridge. (**b**) Schematic of monitoring point location. (**c**) Rover station; (**d**) wind speed sensor; (**e**) wind direction sensor; (**f**) temperature and humidity sensor. (**g**) Reference station.

**Figure 7 sensors-25-03723-f007:**
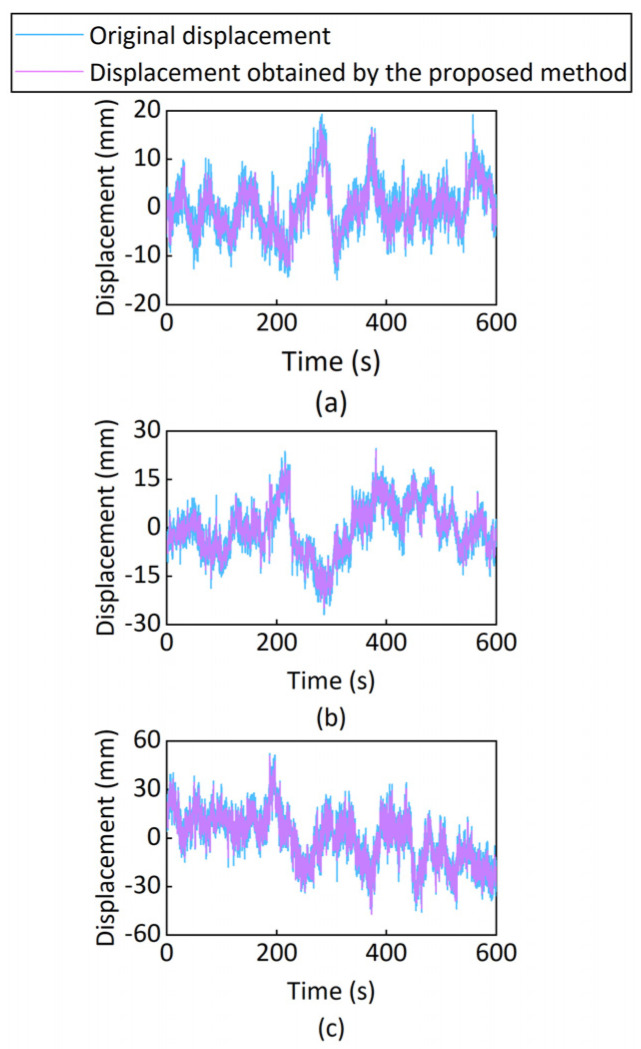
Displacement: (**a**) north–south; (**b**) east–west; (**c**) vertical.

**Figure 8 sensors-25-03723-f008:**
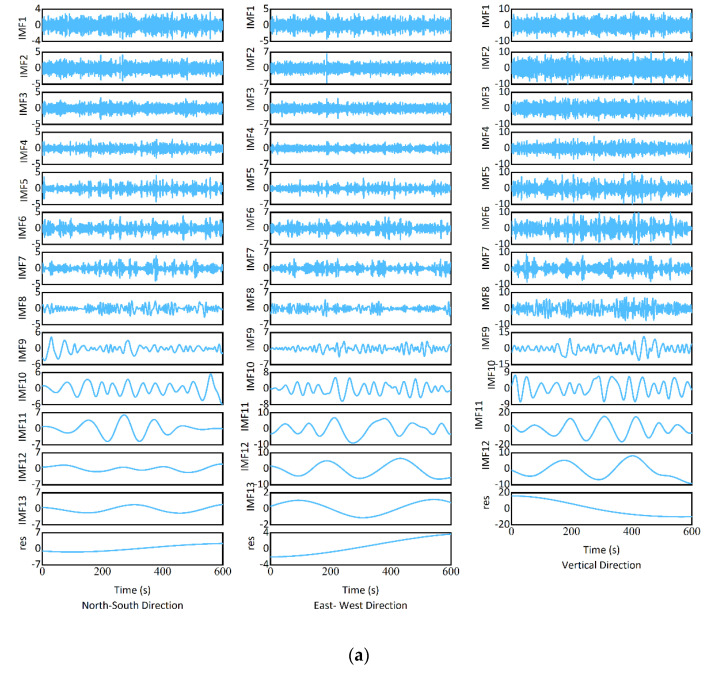
Data process. (**a**) ICEEMDAN decomposition. (**b**) DFA result. (**c**) Signal before and after improved wavelet threshold denoising.

**Figure 9 sensors-25-03723-f009:**
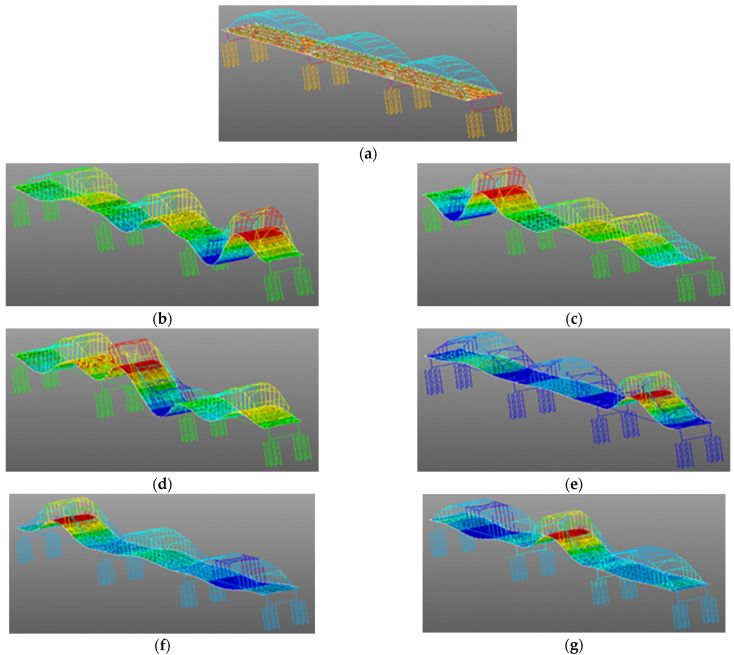
The FEA of Rainbow Bridge. (**a**) Finite element analysis model. (**b**) The first order of the mode shape. (**c**) The second order of the mode shape. (**d**) The third order of the mode shape. (**e**) The fourth order of the mode shape. (**f**) The fifth order of the mode shape. (**g**) The sixth order of the mode shape. (Red, yellow, green, blue and the colors from dark to light indicate the amplitude of the displacement from strong to weak.)

**Figure 10 sensors-25-03723-f010:**
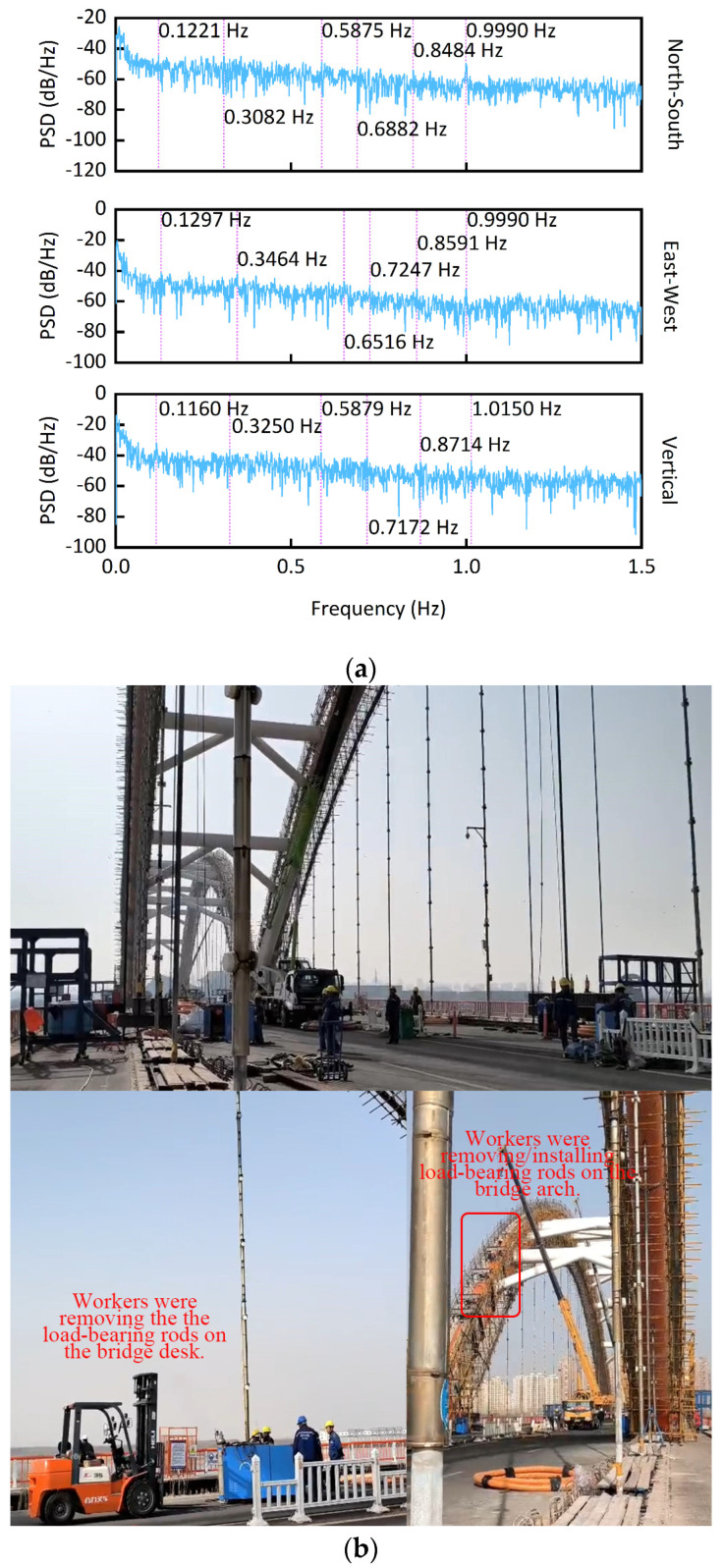
Dynamic characteristics identification and construction site. (**a**) PSD of the monitoring results. (**b**) Construction site.

**Table 1 sensors-25-03723-t001:** Statistical results of SNR and RMSE.

Evaluating Indicators	ICEEMDAN	CEEMDAN-WT	Proposed Method
SNR (dB)	−3.72	−3.08	−2.11
RMSE (mm)	1.02	0.95	0.85

**Table 2 sensors-25-03723-t002:** Displacement range.

Direction		North–South	East–West	Vertical
Original	Displacement (mm)	−14.9–19.3	−26.9–24.7	−46.7–52.3
	Difference (mm)	34.2	51.6	99.0
obtained by the proposed method	Displacement (mm)	−12.3–17.2	−24.6–24.1	−46.7–51.1
	Difference (mm)	29.5	48.7	97.8

## Data Availability

Some or all data, models, or codes that support the findings of this study are available from the corresponding author upon reasonable request.

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
