# Peer review of "Dynamic Monitoring of a Bridge from GNSS-RTK Sensor Using an Improved Hybrid Denoising Method"

_sensors, 2025, doi:10.3390/s25123723_

Round 1

Reviewer 1 Report

Comments and Suggestions for Authors

(1) Abstract: it lacks the introduction of experimental data, such as how long the data were collected, where the data were from, and what the results were like.

(2) Lines 199-200: Figure 6. (a) shows the Rainbow Bridge in maintenance and rehabilitation construction. The field experiment was conducted during the bridge maintenance and rehabilitation construction. These two sentences are repetitive.

(3) Figure 7 is not vivid enough. The original and improved original displacement should be placed in the same figure for comparison.

Comments on the Quality of English Language

No.

Reviewer 2 Report

Comments and Suggestions for Authors

The research paper

‘Dynamic monitoring of a bridge from GNSS-RTK sensor using an improved hybrid denoising method‘

By Xiong et al

Discusses a denoising methodology to improve the accuracy of GNSS-RTK (Global Navigation Satellite System - Real-Time Kinematic) sensors for dynamic displacement monitoring of bridges, especially during maintenance and rehabilitation activities.

The paper is overall interesting, but it is hampered by several limitations, both conceptually and in its editing. Nevertheless, it could be further reconsidered for potential acceptance, if all the following points are properly addressed in the revised version.

  1. With only 15 pages including the Bibliography, the paper is rather short and synthetic. Many theoretical aspects that would require a deeper and more articulated discussion are too briefly mentioned.
  2. The selection of parameters for ICEEMDAN, DFA, and the wavelet function is not discussed. Their impact on the results should be elaborated upon.
  3. The authors should clarify how the threshold λ in Eq. (14) is determined in practice.
  4. The synthetic signal test is useful, but the noise model used (additive Gaussian noise) may not fully represent the real GNSS-RTK error characteristics (e.g., multipath, drift). Consider testing with more realistic noise profiles, as well as potentially sensor fault conditions
  5. More discussion is needed on how the maintenance activities affect dynamic characteristics and the implications for structural safety.
  6. The paper lacks a baseline comparison with other hybrid or advanced denoising methods beyond ICEEMDAN alone.
  7. Related to the above, in the Reviewer’s opinion, the choice of ICEEMDAN, a variant of the well-known CEEMDAN, may be suboptimal. For instance, in “A Comparative Analysis of Signal Decomposition Techniques for Structural Health Monitoring on an Experimental Benchmark”, it was proved that Variational Mode Decomposition, by Dragomiretskiy, Konstantin, and Dominique Zosso. "Variational mode decomposition." IEEE transactions on signal processing 62.3 (2013): 531-544., outperforms significantly other common signal decomposition/denoising alternatives such as CEEMDAN and HVD for SHM-related tasks. This should be included, discussed, and motivated in the text, also referring to the aforementioned comparative study.
  8. Concerning the English, several grammatical and syntactical errors (e.g., missing articles, awkward phrasing like "rapidly from the soft threshold function tends to the hard threshold function", useless repetitions such as “Dynamic characteristics identification revealed the bridge’s features during maintenance and rehabilitation construction. It reminded us to focus on the bridge in maintenance and rehabilitation construction to prevent safety issues caused by excessive mechanical system alterations”, etc) reduce readability and should be corrected by a native speaker or professional editor

Reviewer 3 Report

Comments and Suggestions for Authors

This manuscript presents a methodology for enhancing the accuracy of GNSS-RTK-based monitoring of bridge structures using an improved hybrid denoising algorithm. The proposed method integrates ICEEMDAN decomposition, DFA-based noise identification, and a novel wavelet thresholding strategy. The approach is validated through simulation, a stability test, and a real-world case study involving the Rainbow Bridge in Tianjin, China. The findings demonstrate an improvement in signal-to-noise ratio and reduction of residual noise compared to conventional denoising techniques.

The topic is relevant to the field of structural health monitoring, particularly for infrastructures subject to dynamic loading. The paper addresses a well-recognised limitation of GNSS-RTK in terms of measurement noise and proposes a technically sound and computationally efficient solution. The manuscript is generally well-structured and clearly written, although it would benefit from a thorough language revision.

The technical content is of interest to readers in structural engineering, signal processing, and geodetic monitoring. However, the manuscript would be strengthened by improving the following aspects.

  1. The numerical and experimental validations are effective and support the reliability of the proposed method. However, to further enhance the clarity and generalisability of the results, the authors are encouraged to strengthen the discussion by explicitly highlighting the advantages of the proposed approach in comparison to commonly used denoising strategies. This could be done by referencing relevant literature or discussing, where appropriate, why the chosen method is particularly suited to GNSS-RTK signal characteristics. Additionally, the manuscript would benefit from more detailed information on the implementation of the improved threshold function (e.g., threshold selection criteria, wavelet type, decomposition level), in order to ensure transparency and reproducibility.
  2. The interpretation of the PSD results in Section 4 is somewhat qualitative. The identification of natural frequencies, particularly the lower modes, should be quantified and discussed with reference to expected values and modal shapes, if available from FEA.
  3. Several figures (e.g., Figures 3–9) lack adequate captions or are not referenced in a detailed way in the main text. Additionally, axis labels, units, and legends should be improved to facilitate readability.
  4. The statement “the bridge preserves the structural form envisioned by the designer to some degree” (Section 4) is vague and should be either justified or rephrased
  5. There are some issues concerning the references that should be addressed. Specifically, the DOI in References [4] and [8] appears to be incorrectly formatted or incomplete. Additionally, References [11] and [16] seem to be duplicates, as they report identical author lists and titles but differ in citation style. Please carefully recheck the reference list to ensure accuracy and consistency.

Final Recommendation:

The manuscript presents a meaningful contribution but requires substantial revision in terms of technical comparison, reproducibility, interpretation of results, and language clarity before it can be recommended for publication.

Comments on the Quality of English Language

The manuscript would benefit from a thorough revision of English to improve grammar, syntax, and academic tone. Expressions such as “outperform” (instead of “outperforms”) and informal constructions like “reminded us to” should be corrected for consistency with scientific writing standards. A comprehensive language review is recommended to enhance clarity and readability.

Round 2

Reviewer 2 Report

Comments and Suggestions for Authors

After reviewing the Authors’ revised manuscript and their point-by-point responses, this Reviewer acknowledges the substantial efforts made to address the comments raised in the first review round.

The paper has been extended in length, and the theoretical content has been expanded where previously lacking. Clarifications have been added regarding the selection of key parameters for ICEEMDAN, DFA, and the wavelet thresholding function, including both methodological and practical justifications. The improved noise modeling in the synthetic signal test is also commendable

The authors have also acknowledged that VMD could outperform ICEEMDAN under certain conditions, although the rationale for not adopting VMD remains somewhat limited in explanation. Further elaboration would be beneficial to support their methodological choice more robustly; but apart from that, the content of the paper is acceptable in its current state. n summary, the Authors have addressed the major technical concerns raised during the initial review.

As a last suggestion for proofreading, while the language has improved and several phrases have been revised, the manuscript still contains occasional syntactical and grammatical issues that affect clarity.

Reviewer 3 Report

Comments and Suggestions for Authors

The authors have appropriately revised the article in accordance with the review report. The article can be accepted in its current form.